# Spatiotemporal Dynamics of a Generalized Viral Infection Model with Distributed Delays and CTL Immune Response

**Khalid Hattaf** [1,2]

1   Centre Régional des Métiers de l'Education et de la Formation (CRMEF), 20340 Derb Ghalef, Casablanca, Morocco; k.hattaf@yahoo.fr; Tel.: +212-664407825
2   Laboratory of Analysis, Modeling and Simulation (LAMS), Faculty of Sciences Ben M'sik, Hassan II University, P.O. Box 7955, Sidi Othman, Casablanca, Morocco

**Abstract:** In this paper, we propose and investigate a diffusive viral infection model with distributed delays and cytotoxic T lymphocyte (CTL) immune response. Also, both routes of infection that are virus-to-cell infection and cell-to-cell transmission are modeled by two general nonlinear incidence functions. The well-posedness of the proposed model is also proved by establishing the global existence, uniqueness, nonnegativity and boundedness of solutions. Moreover, the threshold parameters and the global asymptotic stability of equilibria are obtained. Furthermore, diffusive and delayed virus dynamics models presented in many previous studies are improved and generalized.

**Keywords:** viral infection; diffusion; cellular immunity; distributed delay; global stability

## 1. Introduction

During human infections with viruses such as human immunodeficiency virus (HIV), human T-cell leukemia virus (HTLV), hepatitis B virus (HBV) and hepatitis C virus (HCV), cytotoxic T lymphocyte (CTL) cells play a crucial role in antiviral defence by attacking and killing infected cells. So, modeling the role of CTL immune response in viral infection has attracted the attention of many researchers. In 1996, Nowak and Bangham [1] proposed a basic mathematical model by assuming that the infection process is bilinear and follows the principle of mass action. However, as a nonlinear relationship between parasite dose and infection rate has been frequently observed in experiments in [2,3], this bilinear incidence was replaced by Beddington-DeAngelis functional response in [4] and by a more general incidence function in [5].

In the above classical models that are formulated by ordinary differential equations (ODEs), the cell infection is instantaneous and only caused by contact with the free virus. In reality, there are two routes of infection and also time delays in cell infection and virus production. Motivated by these biological reasons, Li et al. [6] proposed a mathematical model formulated by delay differential equations (DDEs) to describe the global dynamics of HIV infection with CTL immune response. This delayed model is an extension of [1] that considers Holling type-II functional response and two kinds of discrete delays, one in cell infection and the other in virus production. Also, the authors of [7] improved the model of Nowak and Bangham [1] by introducing a discrete delay in cell infection and using a Crowley-Martin type incidence function. In 2016, Wang et al. [8] introduced an infinite distributed delay in cell infection in order to improve the basic model with CTL immune response [1], and they also considered both routes of infection, virus-to-cell infection and cell-to-cell transmission. Furthermore, a recent work presented in [9] studied the dynamical behavior of a viral infection model with two types of distributed time delays, CTL immune response and saturated incidence rates for

both routes of infection. In this paper, we generalize all the ODE and DDE models presented in [1,4–9] by proposing the following nonlinear system:

$$
\begin{cases}
\dfrac{dT}{dt} = \lambda - dT(t) - f\big(T(t), I(t), V(t)\big)V(t) - g\big(T(t), I(t)\big)I(t), \\[2mm]
\dfrac{dI}{dt} = \int_0^\infty f_1(\tau)e^{-\alpha_1\tau}[f\big(T(t-\tau), I(t-\tau), V(t-\tau)\big)V(t-\tau) \\[1mm]
\qquad\quad + g\big(T(t-\tau), I(t-\tau)\big)I(t-\tau)]d\tau - aI(t) - pI(t)Z(t), \\[2mm]
\dfrac{dV}{dt} = k\int_0^\infty f_2(\tau)e^{-\alpha_2\tau}I(t-\tau)d\tau - \mu V(t), \\[2mm]
\dfrac{dZ}{dt} = cI(t)Z(t) - bZ(t),
\end{cases}
\tag{1}
$$

where $T(t)$, $I(t)$, $V(t)$ and $Z(t)$ denote the densities of susceptible target cells, infected target cells, free virus particles and CTL cells at time $t$, respectively. The susceptible target cells are produced at constant $\lambda$, die at rate $d$ and become infected by contact with free virus at rate $f(T, I, V)V$ and by contact with infected cells at rate $g(T, I)I$. The parameters $a$ and $b$ are the death rates of infected cells and CTL cells. The parameter $p$ represents the rate at which infected cells are killed by CTL cells, $k$ is the production rate of free virus by an infected cell, and $\mu$ is the clearance rate of free virus. CTL cells expand in response to viral antigens derived from infected cells at rate $cIZ$. Further, we assume that the virus or infected cell contacts an uninfected cell at time $t - \tau$ and the cell becomes infected at time $t$, where $\tau$ is a random variable taken from a probability distribution $f_1(\tau)$. The term $e^{-\alpha_1\tau}$ represents the probability of surviving from time $t - \tau$ to time $t$, where $\alpha_1$ is the death rate for infected but not yet virus-producing cells. In the same, we assume that the time necessary for the newly produced virions to become mature and infectious is a random variable with a probability distribution $f_2(\tau)$. The term $e^{-\alpha_2\tau}$ denotes the probability of the immature virions surviving the delay period, where $\frac{1}{\alpha_2}$ is the average life time of an immature virus. Therefore, the integral $\int_0^\infty f_2(\tau)e^{-\alpha_2\tau}I(t-\tau)d\tau$ describes the mature viral particles produced at time $t$. The probability distribution functions $f_1(\tau)$ and $f_2(\tau)$ are assumed to satisfy $f_i(\tau) \geq 0$ and $\int_0^\infty f_i(\tau)d\tau = 1$ for $i = 1, 2$.

As in [10,11], the incidence functions $f(T, I, V)$ and $g(T, I)$ for both routes of infection are continuously differentiable and satisfy the following hypotheses:

**(H$_0$)** $g(0, I) = 0$, for all $I \geq 0$; $\frac{\partial g}{\partial T}(T, I) \geq 0$ (or $g(T, I)$ is a strictly monotone increasing function with

respect to $T$ when $f \equiv 0$) and $\frac{\partial g}{\partial I}(T, I) \leq 0$, for all $T \geq 0$ and $I \geq 0$.

**(H$_1$)** $f(0, I, V) = 0$, for all $I \geq 0$ and $V \geq 0$,

**(H$_2$)** $f(T, I, V)$ is a strictly monotone increasing function with respect to $T$ (or $\frac{\partial f}{\partial T}(T, I, V) \geq 0$

when $g(T, I)$ is a strictly monotone increasing function with respect to $T$), for any fixed $I \geq 0$ and $V \geq 0$,

**(H$_3$)** $f(T, I, V)$ is a monotone decreasing function with respect to $I$ and $V$.

From a biological viewpoint, the above hypotheses are reasonable and consistent with reality. In fact, the first assumption $(H_0)$ on the function $g(T, I)$ means that the incidence rate by direct contact with infected cells is equal to zero if there are no susceptible cells. This incidence rate is increasing when the number of infected cells is constant and the number of susceptible cells increases. Also, it is decreasing when the number of susceptible cells is constant and the number of infected cells increases. Similarly, the second assumption $(H_1)$ on the function $f(T, I, V)$ means that the incidence rate by contact with free virus is equal to zero if there are no susceptible cells. By $(H_2)$ and $(H_3)$, this incidence rate is increasing when the numbers of infected cells and virus are constant and the number of susceptible cells increases. Also, it is decreasing when the number of susceptible is constant and the number of infected cells or free virus increases. Consequently, the more susceptible cells are, the more infectious events will occur. However, the higher the number of infected cells or the concentration of virus in the host is, the less infectious events will be [10,12,13]. In addition, the functions $f(T, I, V)$

and $g(T, I)$ cover several types of incidence rates existing in the literature such as the classical bilinear incidence, standard incidence, Holling type-II functional response, Beddington-DeAngelis functional response, Crowley-Martin functional response and Hattaf-Yousfi functional response.

On the other hand, system (1) assumes that cells and viruses are well mixed, and ignores their mobility. Actually, viral propagation is a localized process [14] due to the fact that the virus is inherently unstable and the infection occurs mainly in lymphoid tissues. Also, the interaction between virus and the immune response tends to be local within the body of infected hosts [15]. Further, cells are distributed in space and typically interact with the physical environment and other organisms in their spatial neighborhood [16]. Therefore, it is more reasonable to study a reaction-diffusion version of system (1). So, the organization of this paper is as follows. In the next section, we present the reaction-diffusion version of (1) and some preliminary results. Section 3 is devoted to the global dynamics of the reaction-diffusion model. An application and some numerical simulations of our main results are presented in Section 4. Finally, the paper ends with mathematical and biological conclusions in the last section.

## 2. Model Formulation and Preliminaries

We first present a reaction-diffusion version of system (1) by taking into account the mobility of cells and viruses. Hence, system (1) becomes

$$
\begin{cases}
\dfrac{\partial T}{\partial t} = d_T \triangle T + \lambda - dT(x,t) - f\big(T(x,t), I(x,t), V(x,t)\big)V(x,t) \\
\qquad\quad - g\big(T(x,t), I(x,t)\big)I(x,t), \\
\dfrac{\partial I}{\partial t} = d_I \triangle I + \int_0^\infty f_1(\tau)e^{-\alpha_1 \tau}[f\big(T(x,t-\tau), I(x,t-\tau), V(x,t-\tau)\big)V(x,t-\tau) \\
\qquad\quad + g\big(T(x,t-\tau), I(x,t-\tau)\big)I(x,t-\tau)]d\tau - aI(x,t) - pI(x,t)Z(x,t), \\
\dfrac{\partial V}{\partial t} = d_V \triangle V + k\int_0^\infty f_2(\tau)e^{-\alpha_2 \tau}I(x,t-\tau)d\tau - \mu V(x,t), \\
\dfrac{\partial Z}{\partial t} = d_Z \triangle Z + cI(x,t)Z(x,t) - bZ(x,t),
\end{cases}
\tag{2}
$$

where $T(x,t)$, $I(x,t)$, $V(x,t)$ and $Z(x,t)$ are the densities of susceptible target cells, infected target cells, free virus particles and CTL cells at location $x$ and time $t$, respectively. Here, we assume that the motion of the above four populations follows Fickian diffusion, meaning that the fluxes of these four populations are proportional to their concentration gradient and go from regions of high concentration to regions of low concentration, with the diffusion coefficients $d_T$, $d_I$, $d_V$ and $d_Z$, respectively. $\triangle$ is the Laplacian operator. The other parameters have the same biological meanings as those in system (1).

It is very important to note that our model (2) formulated by partial differential equations (PDEs) extends and generalizes many virus dynamics models existing in the literature. For instance, we obtain the diffused HBV infection model proposed by Wang et al. [17] when $d_T = d_I = d_Z = 0$, $f_1(\tau) = f_2(\tau) = \delta(\tau)$, $f(T, I, V) = \beta T^q$ and $g(T, I) = 0$, where $q > 0$, $\beta > 0$ is a constant rate describing the infection process and $\delta(.)$ is the Dirac delta function. When $d_T = d_I = d_Z = 0$, $f_1(\tau) = \delta(\tau - \tau_1)$, $f_2(\tau) = \delta(\tau)$, $f(T, I, V) = \dfrac{\beta T}{1 + \epsilon_1 T + \epsilon_2 V}$ and $g(T, I) = 0$, where $\epsilon_1, \epsilon_2 \geq 0$ are constants, we get the diffusive and delayed viral infection model with Beddington-DeAngelis functional response [18]. Also, the diffusive and delayed viral infection model with Crowley-Martin functional response [19] is a special case of (2), it suffices to take $d_T = d_I = d_Z = 0$, $f_1(\tau) = \delta(\tau - \tau_1)$, $f_2(\tau) = \delta(\tau - \tau_2)$, $f(T, I, V) = \dfrac{\beta T}{(1 + \epsilon_1 T)(1 + \epsilon_2 V)}$ and $g(T, I) = 0$.

Throughout this paper, we consider system (2) with initial conditions

$$
\begin{aligned}
T(x,\theta) &= \phi_1(x,\theta) \geq 0, \quad I(x,\theta) = \phi_2(x,\theta) \geq 0, \\
V(x,\theta) &= \phi_3(x,\theta) \geq 0, \quad Z(x,\theta) = \phi_4(x,\theta) \geq 0, \quad (x,\theta) \in \bar{\Omega} \times (-\infty, 0],
\end{aligned}
\tag{3}
$$

and zero-flux boundary conditions

$$\frac{\partial T}{\partial \nu} = \frac{\partial I}{\partial \nu} = \frac{\partial V}{\partial \nu} = \frac{\partial Z}{\partial \nu} = 0 \ \text{ on } \ \partial\Omega \times (0, +\infty), \tag{4}$$

where $\Omega$ is a bounded domain in $\mathbb{R}^n$ with smooth boundary $\partial\Omega$, and $\dfrac{\partial}{\partial \nu}$ indicates the outward normal derivative on $\partial\Omega$. From the biological point of view, these conditions mean that the uninfected cells, infected cells, free virus particles and CTL cells do not move across the boundary $\partial\Omega$.

We now study the well posedness of the PDE model (2) by establishing the global existence, uniqueness, nonnegativity and boundedness of solutions. To this end, we need some notations. Let $\mathbb{X} = C(\bar{\Omega}, \mathbb{R}^4)$ be the Banach space of continuous functions from $\bar{\Omega}$ into $\mathbb{R}^4$, and $\mathcal{C}_\alpha = C_\alpha((-\infty, 0], \mathbb{X})$ be the Banach space of continuous functions $\varphi$ from $(-\infty, 0]$ into $\mathbb{X}$, where $\varphi(\theta)e^{\alpha\theta}$ is uniformly continuous on $(-\infty, 0]$ and $\|\varphi\| = \sup\limits_{\theta \leq 0}\|\varphi(\theta)\|_{\mathbb{X}}e^{\alpha\theta} < \infty$ with $\alpha$ is a positive constant. For convenience, we identify an element $\varphi \in \mathcal{C}_\alpha$ as a function from $\bar{\Omega} \times (-\infty, 0]$ into $\mathbb{R}^4$ defined by $\varphi(x, \theta) = \varphi(\theta)(x)$. For any continuous function $\omega(.) : (-\infty, \sigma) \to \mathbb{X}$ for $\sigma > 0$, we define $\omega_t \in \mathcal{C}_\alpha$ by $\omega_t(\theta) = \omega(t + \theta)$, $\theta \in (-\infty, 0]$. It is not hard to prove that $t \mapsto \omega_t$ is a continuous function from $[0, \sigma)$ to $\mathcal{C}_\alpha$. Moreover, we need the following lemma.

**Lemma 1.** *Let $A$, $B$ and $D$ be three constants with $B \neq 0$. Consider the following problem*

$$\begin{cases} \dfrac{\partial u}{\partial t} - D\triangle u \leq A - Bu, \ x \in \Omega, \ t > 0, \\[2mm] \dfrac{\partial u}{\partial \nu} = 0, \ \ x \in \partial\Omega, \ t > 0, \\[2mm] u(x, 0) = u_0(x), \ \ x \in \bar{\Omega}. \end{cases} \tag{5}$$

*Then $u(x, t) \leq \max\limits_{x\in\bar{\Omega}} u_0(x)e^{-Bt} + \dfrac{A}{B}(1 - e^{-Bt})$. Moreover, if $B > 0$, we have*

$$u(x, t) \leq \max\left\{\frac{A}{B}, \max\limits_{x\in\bar{\Omega}} u_0(x)\right\} \ \text{ and } \ \limsup\limits_{t\to+\infty} u(x, t) \leq \frac{A}{B}.$$

**Proof.** Let $\tilde{u}(t)$ be a solution to the ordinary differential equation

$$\begin{cases} \dfrac{d\tilde{u}}{dt} = A - B\tilde{u}, \\[2mm] \tilde{u}(0) = \max\limits_{x\in\bar{\Omega}} u_0(x). \end{cases}$$

Then $\tilde{u}(t) = \tilde{u}(0)e^{-Bt} + \frac{A}{B}(1 - e^{-Bt})$. It follows from the comparison principle [20] that $u(x, t) \leq \tilde{u}(t)$. Hence,

$$u(x, t) \leq \max\limits_{x\in\bar{\Omega}} u_0(x)e^{-Bt} + \frac{A}{B}(1 - e^{-Bt}).$$

So, if $B > 0$, we have $u(x, t) \leq \max\left\{\frac{A}{B}, \max\limits_{x\in\bar{\Omega}} u_0(x)\right\}$ and

$$\limsup\limits_{t\to+\infty} u(x, t) \leq \frac{A}{B}.$$

$\square$

**Theorem 1.** *For any given initial condition $\phi \in \mathcal{C}_\alpha$ satisfying (3), problem (2)–(4) has a unique nonnegative solution. When the cells have the same diffusion coefficient ($d_T = d_I = d_Z$), this solution is defined on $[0, +\infty)$ and remains nonnegative and bounded for all $t \geq 0$.*

**Proof.** Let $\varphi = (\varphi_1, \varphi_2, \varphi_3, \varphi_4)^T \in \mathcal{C}_\alpha$ and $x \in \bar{\Omega}$. We define $F = (F_1, F_2, F_3, F_4) : \mathcal{C}_\alpha \to \mathbb{X}$ by

$$
\begin{aligned}
F_1(\varphi)(x) &= \lambda - d\varphi_1(x,0) - f\big(\varphi_1(x,0), \varphi_2(x,0), \varphi_3(x,0)\big)\varphi_3(x,0) \\
&\quad - g\big(\varphi_1(x,0), \varphi_2(x,0)\big)\varphi_2(x,0), \\
F_2(\varphi)(x) &= \int_0^\infty f_1(\tau)e^{-\alpha_1\tau}[f\big(\varphi_1(x,-\tau), \varphi_2(x,-\tau), \varphi_3(x,-\tau)\big)\varphi_3(x,-\tau) \\
&\quad + g\big(\varphi_1(x,-\tau), \varphi_2(x,-\tau)\big)\varphi_2(x,-\tau)]d\tau - a\varphi_2(x,0) - p\varphi_2(x,0)\varphi_4(x,0), \\
F_3(\varphi)(x) &= k\int_0^\infty f_2(\tau)e^{-\alpha_2\tau}\varphi_2(x,-\tau)d\tau - \mu\varphi_3(x,0), \\
F_4(\varphi)(x) &= c\varphi_2(x,0)\varphi_4(x,0) - c\varphi_4(x,0).
\end{aligned}
$$

Then problem (2)–(4) can be rewritten as the following abstract functional differential equation

$$
\begin{cases}
\omega'(t) &= A\omega + F(\omega_t), \quad t > 0, \\
\omega(0) &= \phi \in \mathcal{C}_\alpha,
\end{cases}
\tag{6}
$$

where $\omega = (T, I, V, Z)^T$, $\phi = (\phi_1, \phi_2, \phi_3, \phi_4)^T$ and $A\omega = (d_T \triangle T, d_I \triangle I, d_V \triangle V, d_Z \triangle Z)^T$. It is obvious that $F$ is locally Lipschitz in $\mathcal{C}_\alpha$. According to [21–25], we deduce that system (6) admits a unique local solution on its maximal interval of existence $[0, t_{max})$.

Since $\mathbf{0} = (0,0,0,0)$ is a lower-solution of the problem (2)–(4), we have $T(x,t) \geq 0$, $I(x,t) \geq 0$, $V(x,t) \geq 0$ and $Z(x,t) \geq 0$.

From the first equation of (2), we get

$$
\begin{cases}
\dfrac{\partial T}{\partial t} - d_T \triangle T \leq \lambda - dT, \\
\dfrac{\partial T}{\partial v} = 0, \\
T(x,0) = \phi_1(x,0) \geq 0.
\end{cases}
\tag{7}
$$

By Lemma 1, we get

$$
T(x,t) \leq \max\left\{\frac{\lambda}{d}, \max_{x \in \bar{\Omega}} \phi_1(x,0)\right\}, \forall (x,t) \in \bar{\Omega} \times [0, t_{max}).
$$

This implies that $T$ is bounded. Let

$$
G(x,t) = I(x,t) + \frac{p}{c}Z(x,t) + \int_0^\infty f_1(\tau)e^{-\alpha_1\tau}T(x,t-\tau)d\tau.
$$

The integral in $G(x,t)$ is well-defined and differentiable with respect to $t$, due to $T$ being bounded. Thus,

$$
\begin{aligned}
\frac{\partial G}{\partial t} &= d_T\int_0^\infty f_1(\tau)e^{-\alpha_1\tau}\triangle T(x,t-\tau)d\tau + d_I\triangle I(x,t) + \frac{p}{c}d_Z\triangle Z(x,t) \\
&\quad + \lambda\int_0^\infty f_1(\tau)e^{-\alpha_1\tau}d\tau - d\int_0^\infty f_1(\tau)e^{-\alpha_1\tau}T(x,t-\tau)d\tau - aI(x,t) - \frac{pb}{c}Z(x,t) \\
&\leq d_T\int_0^\infty f_1(\tau)e^{-\alpha_1\tau}\triangle T(x,t-\tau)d\tau + d_I\triangle I(x,t) + \frac{p}{c}d_Z\triangle Z(x,t) + \lambda\eta_1 - \delta G(x,t),
\end{aligned}
$$

where $\delta = \min\{a, b, d\}$ and

$$
\eta_i = \int_0^\infty f_i(\tau)e^{-\alpha_i\tau}d\tau, \quad i = 1,2.
\tag{8}
$$

When $d_T = d_I = d_Z = D$, we have

$$\begin{cases} \dfrac{\partial G}{\partial t} - D\triangle G \leq \lambda\eta_1 - \delta G, \\ \dfrac{\partial G}{\partial \nu} = 0, \\ G(x,0) = \phi_2(x,0) + \dfrac{p}{c}\phi_4(x,0) + \int_0^\infty f_1(\tau)e^{-\alpha_1\tau}\phi_1(0,-\tau)d\tau. \end{cases} \tag{9}$$

From Lemma 1, we have

$$G(x,t) \leq \max\left\{\dfrac{\lambda\eta_1}{\delta}, \max_{x\in\bar\Omega} G(x,0)\right\}, \forall(x,t) \in \bar\Omega \times [0, t_{max}).$$

Thus, $I$ and $Z$ are bounded. It remains to prove that $V$ is bounded. From the boundedness of $I$ and (2)–(4), we deduce that $V$ satisfies the following system

$$\begin{cases} \dfrac{\partial V}{\partial t} - d_V\triangle V \leq kM\eta_2 - \mu V, \\ \dfrac{\partial V}{\partial \nu} = 0, \\ V(x,0) = \phi_3(x,0) \geq 0, \end{cases} \tag{10}$$

where $M = \max\left\{\dfrac{\lambda\eta_1}{\delta}, \max_{x\in\bar\Omega} G(x,0)\right\}$. According to Lemma 1, we deduce that

$$V(x,t) \leq \max\left\{\dfrac{kM\eta_2}{\mu}, \max_{x\in\bar\Omega}\phi_3(x,0)\right\}, \forall(x,t) \in \bar\Omega \times [0, t_{max}).$$

This implies that $V$ is bounded. From the above, we have proved that $T(x,t), I(x,t), V(x,t)$ and $Z(x,t)$ are bounded on $\bar\Omega \times [0, t_{max})$. By the standard theory for semilinear parabolic systems [26], we deduce that $t_{max} = +\infty$. This completes the proof. □

Clearly, system (2) has always one infection-free equilibrium $E_0(T_0, 0, 0, 0)$, where $T_0 = \dfrac{\lambda}{d}$, which represents the healthy state. Hence, we define the basic reproduction number for our PDE model as follows

$$\mathcal{R}_0 = \dfrac{k\eta_1\eta_2 f(\frac{\lambda}{d}, 0, 0) + \mu\eta_1 g(\frac{\lambda}{d}, 0)}{a\mu}. \tag{11}$$

Biologically and as in [11,27], $\mathcal{R}_0$ can be divided into parts as $\mathcal{R}_0 = \mathcal{R}_{01} + \mathcal{R}_{02}$, where $\mathcal{R}_{01} = \dfrac{k\eta_1\eta_2 f(\frac{\lambda}{d}, 0, 0)}{a\mu}$ is the basic reproduction number corresponding to virus-to-cell infection mode, and $\mathcal{R}_{02} = \dfrac{\eta_1 g(\frac{\lambda}{d}, 0)}{a}$ is the basic reproduction number corresponding to cell-to-cell transmission mode.

The other spatially uniform steady states of (2) satisfy the following system

$$\begin{cases} \lambda - dT - f(T,I,V)V - g(T,I)I & = 0, \\ \eta_1\big(f(T,I,V)V + g(T,I)I\big) - aI - pIZ & = 0, \\ k\eta_2 I - \mu V & = 0, \\ cIZ - bZ & = 0. \end{cases} \tag{12}$$

The last equation of (12) implies that $Z = 0$ or $I = \dfrac{b}{c}$. Hence, we discuss two cases. For the case when $Z = 0$, we get

$$k\eta_1\eta_2 f\Big(T, \dfrac{\eta_1(\lambda - dT)}{a}, \dfrac{k\eta_1\eta_2(\lambda - dT)}{a\mu}\Big) + \mu\eta_1 g\Big(T, \dfrac{\eta_1(\lambda - dT)}{a}\Big) = a\mu. \tag{13}$$

Since $I = \dfrac{\eta_1(\lambda - dT)}{a} \geq 0$, we have $T \leq \dfrac{\lambda}{d}$. Then there is no biological equilibrium whenever $T > \dfrac{\lambda}{d}$. Let us define the function $\psi_1$ on the interval $[0, \frac{\lambda}{d}]$ by

$$\psi_1(T) \quad = \quad k\eta_1\eta_2 f\Big(T, \frac{\eta_1(\lambda - dT)}{a}, \frac{k\eta_1\eta_2(\lambda - dT)}{a\mu}\Big) + \mu\eta_1 g\Big(T, \frac{\eta_1(\lambda - dT)}{a}\Big) - a\mu.$$

It follows from $(H_0)$–$(H_3)$ that $\psi_1(0) = -a\mu < 0$, $\psi_1(\frac{\lambda}{d}) = a\mu(\mathcal{R}_0 - 1)$ and

$$\psi_1'(T) = k\eta_1\eta_2\Big(\frac{\partial f}{\partial T} - \frac{d\eta_1}{a}\frac{\partial f}{\partial I} - \frac{kd\eta_1\eta_2}{a\mu}\frac{\partial f}{\partial V}\Big) + \mu\eta_1\Big(\frac{\partial g}{\partial T} - \frac{d\eta_1}{a}\frac{\partial g}{\partial I}\Big) > 0,$$

which implies that there exists a unique $T_1 \in (0, \frac{\lambda}{d})$ such as $\psi_1(T_1) = 0$ provided that $\mathcal{R}_0 > 1$. Thus, $E_1(T_1, I_1, V_1, 0)$ is a unique infection equilibrium of (2) with $I_1 = \dfrac{\eta_1(\lambda - dT_1)}{a}$ and $V_1 = \dfrac{k\eta_1\eta_2(\lambda - dT_1)}{a\mu}$.

For the case when $Z \neq 0$, we have $I = \dfrac{b}{c}$, $V = \dfrac{kb\eta_2}{c\mu}$ and

$$k\eta_2 f\Big(T, \frac{b}{c}, \frac{kb\eta_2}{c\mu}\Big) + \mu g\Big(T, \frac{b}{c}\Big) = \frac{c\mu}{b}(\lambda - dT). \tag{14}$$

Since $Z = \dfrac{c\eta_1(\lambda - dT) - ab}{pb} \geq 0$, we have $T \leq \dfrac{\lambda}{d} - \dfrac{ab}{dc\eta_1}$. Then there is no positive equilibrium when $T > \dfrac{\lambda}{d} - \dfrac{ab}{dc\eta_1}$ or $\dfrac{\lambda}{d} - \dfrac{ab}{dc\eta_1} \leq 0$. Define the function $\psi_2$ on the interval $[0, \frac{\lambda}{d} - \frac{ab}{dc\eta_1}]$ by

$$\psi_2(T) \quad = \quad k\eta_2 f\Big(T, \frac{b}{c}, \frac{kb\eta_2}{c\mu}\Big) + \mu g\Big(T, \frac{b}{c}\Big) - \frac{c\mu}{b}(\lambda - dT).$$

If CTL immune response has not been established, we have $cI_1 - b \leq 0$. So, we define the reproduction number for cellular immunity as follows

$$\mathcal{R}_1^Z = \frac{cI_1}{b}, \tag{15}$$

where $\dfrac{1}{b}$ denotes the average life expectancy of CTL cells and $I_1$ is the number of infected cells at $E_1$. Hence, $\mathcal{R}_1^Z$ represents the average number of the CTL immune cells activated by infected cells. If $\mathcal{R}_1^Z < 1$, then $I_1 < \dfrac{c}{b}$, $T_1 > \dfrac{\lambda}{d} - \dfrac{ab}{dc\eta_1}$ and

$$\psi_2\Big(\frac{\lambda}{d} - \frac{ab}{dc\eta_1}\Big) = \frac{1}{\eta_1}\psi_1\Big(\frac{\lambda}{d} - \frac{ab}{dc\eta_1}\Big) < \frac{1}{\eta_1}\psi_1(T_1) = 0.$$

So, there is no equilibrium when $\mathcal{R}_1^Z < 1$.

If $\mathcal{R}_1^Z > 1$, then $I_1 > \dfrac{c}{b}$, $T_1 < \dfrac{\lambda}{d} - \dfrac{ab}{dc\eta_1}$ and $\psi_2\Big(\frac{\lambda}{d} - \frac{ab}{dc\eta_1}\Big) > 0$. Hence, if $\mathcal{R}_1^Z > 1$, system (2) has a CTL-activated infection equilibrium $E_2(T_2, I_2, V_2, Z_2)$, where $T_2 \in (0, \frac{\lambda}{d} - \frac{ab}{dc\eta_1})$, $I_2 = \dfrac{b}{c}$, $V_2 = \dfrac{kb\eta_2}{\mu c}$ and $Z_2 = \dfrac{c\eta_1(\lambda - dT_2) - ab}{pb}$.

Recapitulating the above discussions in the following theorem.

**Theorem 2.**

(i) If $\mathcal{R}_0 \leq 1$, then system (2) has a unique infection-free equilibrium $E_0(T_0, 0, 0, 0)$, where $T_0 = \dfrac{\lambda}{d}$.

(ii) If $\mathcal{R}_0 > 1$, then system (2) has a unique infection equilibrium without cellular immunity $E_1(T_1, I_1, V_1, 0)$ besides $E_0$, where $T_1 \in (0, \dfrac{\lambda}{d})$, $I_1 = \dfrac{\eta_1(\lambda - dT_1)}{a}$ and $V_1 = \dfrac{k\eta_1\eta_2(\lambda - dT_1)}{a\mu}$.

(iii) If $\mathcal{R}_1^Z > 1$, then system (2) has a unique infection equilibrium with cellular immunity $E_2(T_2, I_2, V_2, Z_2)$ besides $E_0$ and $E_1$, where $T_2 \in (0, \dfrac{\lambda}{d} - \dfrac{ab}{dc\eta_1})$, $I_2 = \dfrac{b}{c}$, $V_2 = \dfrac{kb\eta_2}{\mu c}$ and $Z_2 = \dfrac{c\eta_1(\lambda - dT_2) - ab}{pb}$.

## 3. Global Stability

Regarding the global stability of the infection-free equilibrium $E_0$, we have the following theorem.

**Theorem 3.** *The infection-free equilibrium $E_0$ of system (2) is globally asymptotically stable when $\mathcal{R}_0 \leq 1$.*

**Proof.** Based on the method proposed in [28], we construct the Lyapunov functional for system (2) at $E_0$ as follows

$$
\begin{aligned}
L_0 =\ & \int_\Omega \left\{ \frac{1}{\eta_1} I(x,t) + \frac{f(T_0,0,0)}{\mu} V(x,t) + \frac{p}{c\eta_1} Z(x,t) \right. \\
& + \frac{1}{\eta_1} \int_0^\infty f_1(\tau) e^{-\alpha_1 \tau} \int_{t-\tau}^t \left[ f(T(x,s), I(x,s), V(x,s)) V(x,s) \right. \\
& \left. + g(T(x,s), I(x,s)) I(x,s) \right] ds d\tau + \frac{kf(T_0,0,0)}{\mu} \int_0^\infty h_2(\tau) e^{-\alpha_2 \tau} \int_{t-\tau}^t I(x,s) ds d\tau \bigg\} dx.
\end{aligned}
$$

For convenience, we let $\varphi = \varphi(x,t)$ and $\varphi_\tau = \varphi(x, t-\tau)$ for any $\varphi \in \{u, w, v, z\}$. The time derivative of $L_0$ along the solution of system (2) satisfies

$$
\begin{aligned}
\frac{dL_0}{dt} =\ & \int_\Omega \left\{ \left( f(T, I, V) - f(T_0, 0, 0) \right) V + \frac{a}{\eta_1} I \left( \frac{k\eta_1\eta_2 f(T_0,0,0) + \mu\eta_1 g(T,I)}{a\mu} - 1 \right) \right. \\
& - \frac{pb}{c\eta_1} Z \bigg\} dx.
\end{aligned}
$$

From (7) and by applying Lemma 1, we get $\limsup\limits_{t \to \infty} T(x,t) \leq T_0$. This implies that all omega limit points satisfy $T(x,t) \leq T_0$. Hence, it is sufficient to consider solutions for which $T(x,t) \leq T_0$. From the explicit formula of $\mathcal{R}_0$ given in (11) and $(H_0)$-$(H_3)$, we get

$$
\begin{aligned}
\frac{dL_0}{dt} & \leq \int_\Omega \left\{ \left( f(T, I, V) - f(T_0, 0, 0) \right) V + \frac{a}{\eta_1} (\mathcal{R}_0 - 1) I - \frac{pb}{c\eta_1} Z \right\} dx \\
& \leq \int_\Omega \left\{ \frac{a}{\eta_1} (\mathcal{R}_0 - 1) I - \frac{pb}{c\eta_1} Z \right\} dx.
\end{aligned}
$$

Hence, $\mathcal{R}_0 \leq 1$ ensures $\dfrac{dL_0}{dt} \leq 0$. In addition, it can be shown that the largest compact invariant set in $\{(T, I, V, Z) | \dfrac{dL_0}{dt} = 0\}$ is the singleton $\{E_0\}$. Therefore, it follows from LaSalle's invariance principle [29] that $E_0$ is globally asymptotically stable when $\mathcal{R}_0 \leq 1$. $\square$

For the global stability of the two infection steady states $E_i$ of system (2), we suppose that $\mathcal{R}_0 > 1$ and the incidence functions $f$ and $g$ satisfy for each infection equilibrium $E_i$ the following further hypothesis

$$\left(1 - \frac{f(T, I, V)}{f(T, I_i, V_i)}\right)\left(\frac{f(T, I_i, V_i)}{f(T, I, V)} - \frac{V}{V_i}\right) \leq 0,$$
$$\left(1 - \frac{f(T_i, I_i, V_i)g(T, I)}{f(T, I_i, V_i)g(T_i, I_i)}\right)\left(\frac{f(T, I_i, V_i)g(T_i, I_i)}{f(T_i, I_i, V_i)g(T, I)} - \frac{I}{I_i}\right) \leq 0. \tag{$H_4$}$$

Therefore, we have the following result.

**Theorem 4.** *Assume* $\mathcal{R}_0 > 1$ *and* ($H_4$) *holds for each* $E_i$.

**(i)** *The infection equilibrium without cellular immunity* $E_1$ *of system* (2) *is globally asymptotically stable if* $\mathcal{R}_1^z \leq 1$.

**(ii)** *The infection equilibrium with cellular immunity* $E_2$ *of system* (2) *is globally asymptotically stable if* $\mathcal{R}_1^z > 1$.

**Proof.** For **(i)**, we construct the Lyapunov functional as follows

$$\begin{aligned}
L_1 &= \int_\Omega \left\{ T - T_1 - \int_{T_1}^T \frac{f(T_1, I_1, V_1)}{f(X, I_1, V_1)} dX + \frac{1}{\eta_1} I_1 \Phi\left(\frac{I}{I_1}\right) \right. \\
&\quad + \frac{f(T_1, I_1, V_1)V_1}{k\eta_2 I_1} V_1 \Phi\left(\frac{V}{V_1}\right) + \frac{p}{c\eta_1} Z \\
&\quad + \frac{1}{\eta_1} f(T_1, I_1, V_1)V_1 \int_0^\infty f_1(\tau)e^{-\alpha_1\tau} \int_{t-\tau}^t \Phi\left(\frac{f(T(x,s), I(x,s), V(x,s))V(x,s)}{f(T_1, I_1, V_1)V_1}\right) ds d\tau \\
&\quad + \frac{1}{\eta_1} g(T_1, I_1)I_1 \int_0^\infty f_1(\tau)e^{-\alpha_1\tau} \int_{t-\tau}^t \Phi\left(\frac{g(T(x,s), I(x,s))I(x,s)}{g(T_1, I_1)I_1}\right) ds d\tau \\
&\quad \left. + \frac{1}{\eta_2} f(T_1, I_1, V_1)V_1 \int_0^\infty f_2(\tau)e^{-\alpha_2\tau} \int_{t-\tau}^t \Phi\left(\frac{I(x,s)}{I_1}\right) ds d\tau \right\} dx,
\end{aligned}$$

where $\Phi(\xi) = \xi - 1 - \ln\xi$, $\xi > 0$. Clearly, $\Phi : (0, +\infty) \to [0, +\infty)$ attains its strict global minimum at $\xi = 1$ and $\Phi(1) = 0$. Then $\Phi(\xi) \geq 0$ and so the functional $L_1$ is non-negative. Calculating the time derivative of $L_1$ along the solution of system (2), we obtain

$$\begin{aligned}
\frac{dL_1}{dt} &= \int_\Omega \left\{ \left(1 - \frac{f(T_1, I_1, V_1)}{f(T, I_1, V_1)}\right)\frac{\partial T}{\partial t} + \frac{1}{\eta_1}\left(1 - \frac{I_1}{I}\right)\frac{\partial I}{\partial t} \right. \\
&\quad + \frac{f(T_1, I_1, V_1)V_1}{k\eta_2 I_1}\left(1 - \frac{V_1}{V}\right)\frac{\partial V}{\partial t} + \frac{p}{c\eta_1}\frac{\partial Z}{\partial t} \\
&\quad + \frac{1}{\eta_1} f(T_1, I_1, V_1)V_1 \int_0^\infty f_1(\tau)e^{-\alpha_1\tau}\left(\Phi\left(\frac{f(T, I, V)V}{f(T_1, I_1, V_1)V_1}\right) - \Phi\left(\frac{f(T_\tau, I_\tau, V_\tau)V_\tau}{f(T_1, I_1, V_1)V_1}\right)\right) d\tau \\
&\quad + \frac{1}{\eta_1} g(T_1, I_1)I_1 \int_0^\infty f_1(\tau)e^{-\alpha_1\tau}\left(\Phi\left(\frac{g(T, I)I}{g(T_1, I_1)I_1}\right) - \Phi\left(\frac{g(T_\tau, I_\tau)I_\tau}{g(T_1, I_1)I_1}\right)\right) d\tau \\
&\quad \left. + \frac{1}{\eta_2} f(T_1, I_1, V_1)V_1 \int_0^\infty f_2(\tau)e^{-\alpha_2\tau}\left(\Phi\left(\frac{I}{I_1}\right) - \Phi\left(\frac{I_\tau}{I_1}\right)\right) d\tau \right\} dx.
\end{aligned}$$

Using $\lambda = dT_1 + f(T_1, I_1, V_1)V_1 + g(T_1, I_1)I_1 = dT_1 + \dfrac{a}{\eta_1}I_1$ and $k\eta_2 I_1 = \mu V_1$, we get

$$
\begin{aligned}
\frac{dL_1}{dt} &= \int_\Omega \Bigg\{ dT_1\left(1 - \frac{T}{T_1}\right)\left(1 - \frac{f(T_1, I_1, V_1)}{f(T, I_1, V_1)}\right) \\
&\quad + f(T_1, I_1, V_1)V_1\left(-1 - \frac{V}{V_1} + \frac{f(T, I_1, V_1)}{f(T, I, V)} + \frac{f(T, I, V)V}{f(T, I_1, V_1)V_1}\right) \\
&\quad + g(T_1, I_1)I_1\left(-1 - \frac{I}{I_1} + \frac{f(T, I_1, V_1)g(T_1, I_1)}{f(T, I_1, V_1)g(T, I)} + \frac{f(T_1, I_1, V_1)g(T, I)I}{f(T, I_1, V_1)g(T_1, I_1)I_1}\right) \\
&\quad - \frac{1}{\eta_1}f(T_1, I_1, V_1)V_1\int_0^\infty f_1(\tau)e^{-\alpha_1\tau}\left[\Phi\left(\frac{f(T_1, I_1, V_1)}{f(T, I_1, V_1)}\right) + \Phi\left(\frac{f(T_\tau, I_\tau, V_\tau)V_\tau I_1}{f(T_1, I_1, V_1)V_1 I}\right)\right. \\
&\quad \left. + \Phi\left(\frac{f(T, I_1, V_1)}{f(T, I, V)}\right)\right]d\tau - \frac{1}{\eta_1}g(T_1, I_1)I_1\int_0^\infty f_1(\tau)e^{-\alpha_1\tau}\left[\Phi\left(\frac{f(T_1, I_1, V_1)}{f(T, I_1, V_1)}\right)\right. \\
&\quad \left. + \Phi\left(\frac{g(T_\tau, I_\tau)I_\tau}{g(T_1, I_1)I}\right) + \Phi\left(\frac{f(T, I_1, V_1)g(T_1, I_1)}{f(T_1, I_1, V_1)g(T, I)}\right)\right]d\tau \\
&\quad - \frac{1}{\eta_2}f(T_1, I_1, V_1)V_1\int_0^\infty f_2(\tau)e^{-\alpha_2\tau}\Phi\left(\frac{V_1 I_\tau}{V I_1}\right)d\tau + \frac{pb}{c\eta_1}(\mathcal{R}_1^Z - 1)Z\Bigg\}dx \\
&\quad - f(T_1, I_1, V_1)d_T\int_\Omega \frac{\partial f}{\partial T}(T, I_1, V_1)\frac{|\nabla T|^2}{[f(T, I_1, V_1)]^2}dx \\
&\quad - \frac{I_1 d_I}{\eta_1}\int_\Omega \frac{|\nabla I|^2}{I^2}dx - \frac{f(T_1, I_1, V_1)V_1}{\mu}d_V\int_\Omega \frac{|\nabla I|^2}{I^2}dx.
\end{aligned}
$$

Since the function $f(T, I, V)$ is strictly monotonically increasing with respect to $T$, we have for $i = 1, 2$ that

$$
\frac{\partial f}{\partial T}(T, I_i, V_i) > 0 \text{ and } \left(1 - \frac{T}{T_i}\right)\left(1 - \frac{f(T_i, I_i, V_i)}{f(T, I_i, V_i)}\right) \le 0. \tag{16}
$$

It follows from ($H_4$) that

$$
-1 - \frac{V}{V_i} + \frac{f(T, I_i, V_i)}{f(T, I, V)} + \frac{f(T, I, V)V}{f(T, I_i, V_i)V_i} = \left(1 - \frac{f(T, I, V)}{f(T, I_i, V_i)}\right)\left(\frac{f(T, I_i, V_i)}{f(T, I, V)} - \frac{V}{V_i}\right) \le 0, \tag{17}
$$

and

$$
\begin{aligned}
&-1 - \frac{I}{I_1} + \frac{f(T, I_i, V_i)g(T_i, I_i)}{f(T_i, I_i, V_i)g(T, I)} + \frac{f(T_i, I_i, V_i)g(T, I)I}{f(T, I_i, V_i)g(T_i, I_i)I_i} \\
&= \left(1 - \frac{f(T_i, I_i, V_i)g(T, I)}{f(T, I_i, V_i)g(T_i, I_i)}\right)\left(\frac{f(T, I_i, V_i)g(T_i, I_i)}{f(T_i, I_i, V_i)g(T, I)} - \frac{I}{I_i}\right) \le 0.
\end{aligned} \tag{18}
$$

Since $H(\xi) \ge 0$ and $\mathcal{R}_1^Z \le 1$, we have $\dfrac{dL_1}{dt} \le 0$ with equality if and only if $T = T_1$, $I = I_1$, $V = V_1$ and $Z = 0$. It follows from LaSalle's invariance principle that $E_1$ is globally asymptotically stable.

For **(ii)**, we construct the Lyapunov functional as follows

$$
\begin{aligned}
L_2 &= \int_\Omega \Bigg\{ T - T_2 - \int_{T_2}^T \frac{f(T_2, I_2, V_2)}{f(X, I_2, V_2)}dX + \frac{1}{\eta_1}I_2\Phi\left(\frac{I}{I_2}\right) \\
&\quad + \frac{f(T_2, I_2, V_2)V_2}{k\eta_2 I_2}V_2\Phi\left(\frac{V}{V_2}\right) + \frac{p}{c\eta_1}Z_2\Phi\left(\frac{Z}{Z_2}\right) \\
&\quad + \frac{1}{\eta_1}f(T_2, I_2, V_2)V_2\int_0^\infty f_1(\tau)e^{-\alpha_1\tau}\int_{t-\tau}^t \Phi\left(\frac{f(T(x,s), I(x,s), V(x,s))V(x,s)}{f(T_2, I_2, V_2)V_2}\right)dsd\tau \\
&\quad + \frac{1}{\eta_1}g(T_2, I_2)I_2\int_0^\infty f_1(\tau)e^{-\alpha_1\tau}\int_{t-\tau}^t \Phi\left(\frac{g(T(x,s), I(x,s))I(x,s)}{g(T_2, I_2)I_2}\right)dsd\tau \\
&\quad + \frac{1}{\eta_2}f(T_2, I_2, V_2)V_2\int_0^\infty f_2(\tau)e^{-\alpha_2\tau}\int_{t-\tau}^t \Phi\left(\frac{I(x,s)}{I_2}\right)dsd\tau \Bigg\}dx.
\end{aligned}
$$

Calculating the time derivative of $L_2$ along the solution of system (2) and using $\lambda = dT_2 + f(T_2, I_2, V_2)V_2 + g(T_2, I_2)I_2 = dT_2 + \dfrac{a}{\eta_1}I_2 + \dfrac{p}{\eta_1}I_2Z_2$, $I_2 = \dfrac{b}{c}$ and $k\eta_2 I_2 = \mu V_2$, we have

$$
\begin{aligned}
\frac{dL_2}{dt} = & \int_\Omega \Bigg\{ dT_2\left(1 - \frac{T}{T_2}\right)\left(1 - \frac{f(T_2, I_2, V_2)}{f(T, I_2, V_2)}\right) \\
& + f(T_2, I_2, V_2)V_2\left(-1 - \frac{V}{V_2} + \frac{f(T, I_2, V_2)}{f(T, I, V)} + \frac{f(T, I, V)V}{f(T, I_2, V_2)V_2}\right) \\
& + g(T_2, I_2)I_2\left(-1 - \frac{I}{I_2} + \frac{f(T, I_2, V_2)g(T_2, I_2)}{f(T_2, I_2, V_2)g(T, I)} + \frac{f(T_2, I_2, V_2)g(T, I)I}{f(T, I_2, V_2)g(T_2, I_2)I_2}\right) \\
& - \frac{1}{\eta_1}f(T_2, I_2, V_2)V_2\int_0^\infty f_1(\tau)e^{-\alpha_1\tau}\left[\Phi\left(\frac{f(T_2, I_2, V_2)}{f(T, I_2, V_2)}\right) + \Phi\left(\frac{f(T_\tau, I_\tau, V_\tau)V_\tau I_2}{f(T_2, I_2, V_2)V_2 I}\right)\right. \\
& \left. + \Phi\left(\frac{f(T, I_2, V_2)}{f(T, I, V)}\right)\right]d\tau - \frac{1}{\eta_1}g(T_2, I_2)I_2\int_0^\infty f_1(\tau)e^{-\alpha_1\tau}\left[\Phi\left(\frac{f(T_2, I_2, V_2)}{f(T, I_2, V_2)}\right)\right. \\
& \left. + \Phi\left(\frac{g(T_\tau, I_\tau)I_\tau}{g(T_1, I_1)I}\right) + \Phi\left(\frac{f(T, I_2, V_2)g(T_2, I_2)}{f(T_2, I_2, V_2)g(T, I)}\right)\right]d\tau \\
& - \frac{1}{\eta_2}f(T_2, I_2, V_2)V_2\int_0^\infty f_2(\tau)e^{-\alpha_2\tau}\Phi\left(\frac{V_2 I_\tau}{V I_2}\right)d\tau\Bigg\}dx \\
& - f(T_2, I_2, V_2)d_T\int_\Omega \frac{\partial f}{\partial T}(T, I_2, V_2)\frac{|\nabla T|^2}{[f(T, I_2, V_2)]^2}dx \\
& - \frac{I_2 d_I}{\eta_1}\int_\Omega \frac{|\nabla I|^2}{I^2}dx - \frac{f(T_2, I_2, V_2)V_2}{\mu}d_V\int_\Omega \frac{|\nabla I|^2}{I^2}dx.
\end{aligned}
$$

From (16)–(18), we have $\dfrac{dL_2}{dt} \leq 0$. Further, it is not hard to see that the largest invariant set in $\{(T, I, V, Z)|\dfrac{dL_2}{dt} = 0\}$ is $\{E_2\}$. By LaSalle's invariance principle, we deduce that $E_2$ is globally asymptotically stable. This ends the proof of Theorem 4. $\quad\square$

**Remark 1.** *The hypothesis ($H_4$) comes from (17) and (18). This hypothesis is a sufficient condition for that the time derivatives of the Lyapunov functionals $L_1$ and $L_2$ to be non-negative. When cell-to-cell mode is ignored (i.e., $g \equiv 0$), the assumption ($H_4$) can be reduced to*

$$
\left(1 - \frac{f(T, I, V)}{f(T, I_i, V_i)}\right)\left(\frac{f(T, I_i, V_i)}{f(T, I, V)} - \frac{V}{V_i}\right) \leq 0, \tag{$H_4'$}
$$

*which is verified by many types of the incidence rate including the bilinear incidence, the saturation incidence, the Beddington-DeAnglis functional response, the Crowley-Martin functional response and the Hattaf-Yousfi functional response.*

In 2017, Xu et al. [30] proposed a PDE model with two discrete delays, cell-to-cell transmission and CTL immune response. They considered the spatial diffusion only in virus. This model is given by

$$
\begin{cases}
\dfrac{\partial T}{\partial t} = \lambda - dT(x, t) - \beta_1 T(x, t)\tilde{f}(V(x, t)) - \beta_2 T(x, t)\tilde{g}(I(x, t)), \\
\dfrac{\partial I}{\partial t} = \beta_1 T(x, t - \tau_1)\tilde{f}(V(x, t - \tau_1)) + \beta_2 T(x, t - \tau_1)\tilde{g}(I(x, t - \tau_1)) \\
\qquad - aI(x, t) - pI(x, t)Z(x, t), \\
\dfrac{\partial V}{\partial t} = d_V \triangle V + kI(x, t - \tau_2) - \mu V(x, t), \\
\dfrac{\partial Z}{\partial t} = cI(x, t)Z(x, t) - bZ(x, t),
\end{cases} \tag{19}
$$

where the functions $\tilde{f}$ and $\tilde{g}$ satisfy the following properties:

$$\tilde{f}(0) = \tilde{g}(0) = 0, \quad \tilde{f}'(V) > 0, \quad \tilde{g}'(I) > 0, \quad \tilde{f}''(V) \le 0, \quad \tilde{g}''(I) \le 0. \tag{20}$$

Choose the functions $f(T, I, V)$ and $g(T, I)$ as follows

$$f(T, I, V) = \begin{cases} \frac{\beta_1 T \tilde{f}(V)}{V}, & V \ne 0, \\ \beta_1 T \tilde{f}'(0), & V = 0, \end{cases} \quad \text{and} \quad g(T, I) = \begin{cases} \frac{\beta_2 T \tilde{g}(I)}{I}, & I \ne 0, \\ \beta_2 T \tilde{g}'(0), & I = 0. \end{cases}$$

Clearly, $f(T, I, V)V = \beta_1 T \tilde{f}(V)$ and $g(T, I)I = \beta_2 T \tilde{g}(I)$ for all $T, I, V \ge 0$. Based on $\left(\frac{\tilde{f}(V)}{V}\right)' \le 0$, $\left(\frac{\tilde{g}(I)}{I}\right)' \le 0$ and the last inequality of Lemma 3.1 in [30], it is not hard to prove that the above incidence functions $f(T, I, V)$ and $g(T, I)$ satisfy the five hypotheses $(H_0) - (H_4)$. Therefore, the model and results investigated in [30] are extended and generalized.

## 4. Application and Numerical Simulations

In this section, we first apply our main results obtained in this study to the following model:

$$\begin{cases} \frac{\partial T}{\partial t} = d_T \triangle T + \lambda - dT(x, t) - \frac{\beta_1 T(x, t)V(x, t)}{1 + \epsilon_1 V(x, t)} - \frac{\beta_2 T(x, t)I(x, t)}{1 + \epsilon_2 I(x, t)}, \\ \frac{\partial I}{\partial t} = d_I \triangle I + \int_0^\infty f_1(\tau)e^{-\alpha_1 \tau}\left[\frac{\beta_1 T(x, t - \tau)V(x, t - \tau)}{1 + \epsilon_1 V(x, t - \tau)} + \frac{\beta_2 T(x, t - \tau)I(x, t - \tau)}{1 + \epsilon_2 I(x, t - \tau)}\right]d\tau \\ \qquad -aI(x, t) - pI(x, t)Z(x, t), \\ \frac{\partial V}{\partial t} = d_V \triangle V + k\int_0^\infty f_2(\tau)e^{-\alpha_2 \tau}I(x, t - \tau)d\tau - \mu V(x, t), \\ \frac{\partial Z}{\partial t} = d_Z \triangle Z + cI(x, t)Z(x, t) - bZ(x, t), \end{cases} \tag{21}$$

where $\beta_1$ and $\beta_2$ denote, respectively, the virus-to-cell infection rate and the cell-to-cell transmission rate. The non-negative constants $\epsilon_1$ and $\epsilon_2$ measure the saturation effect. The other state variables and parameters have the same biological meanings as in models (1) and (2). Notice that system (21) extends the DDE model presented in [9] by introducing the spacial diffusion in both cells and viruses. Also, this system is a particular case of (2) with $f(T, I, V) = \frac{\beta_1 T}{1 + \epsilon_1 V}$ and $g(I, I) = \frac{\beta_2 T}{1 + \epsilon_2 I}$. As before, we consider system (21) with initial conditions

$$\begin{aligned} T(x, \theta) &= \phi_1(x, \theta) \ge 0, \quad I(x, \theta) = \phi_2(x, \theta) \ge 0, \\ V(x, \theta) &= \phi_3(x, \theta) \ge 0, \quad Z(x, \theta) = \phi_4(x, \theta) \ge 0, \quad (x, \theta) \in \bar{\Omega} \times (-\infty, 0], \end{aligned} \tag{22}$$

and Neumann boundary conditions

$$\frac{\partial T}{\partial \nu} = \frac{\partial I}{\partial \nu} = \frac{\partial V}{\partial \nu} = \frac{\partial Z}{\partial \nu} = 0 \quad \text{on} \quad \partial\Omega \times (0, +\infty). \tag{23}$$

It is easy to check the first four hypotheses $(H_0)$-$(H_3)$. For the fifth hypothesis, we have

$$\left(1 - \frac{f(T, I, V)}{f(T, I_i, V_i)}\right)\left(\frac{f(T, I_i, V_i)}{f(T, I, V)} - \frac{V}{V_i}\right) = \frac{-\epsilon_1(V - V_i)^2}{V_i(1 + \epsilon_1 V_i)(1 + \epsilon_1 V)} \le 0,$$

$$\left(1 - \frac{f(T_i, I_i, V_i)g(T, I)}{f(T, I_i, V_i)g(T_i, I_i)}\right)\left(\frac{f(T, I_i, V_i)g(T_i, I_i)}{f(T_i, I_i, V_i)g(T, I)} - \frac{I}{I_i}\right) = \frac{-\epsilon_2(I - I_i)^2}{I_i(1 + \epsilon_2 I_i)(1 + \epsilon_2 I)} \le 0.$$

Thus, the last hypothesis $(H_4)$ is verified. By applying Theorems 3 and 4, we obtain the following result.

**Corollary 1.**

1.  *If $\mathcal{R}_0 \leq 1$, then the infection-free equilibrium $E_0$ of system (21) is globally asymptotically stable.*
2.  *If $\mathcal{R}_0 > 1$, then system (21) has two infection equilibria that are:*

    **(i)**   *the infection equilibrium without cellular immunity $E_1$ that is globally asymptotically stable if $\mathcal{R}_1^Z \leq 1$;*
    **(ii)**  *the infection equilibrium with cellular immunity $E_2$ that is globally asymptotically stable if $\mathcal{R}_1^Z > 1$.*

For the numerical simulations, we choose $f_i(\tau) = \gamma_i e^{-\gamma_i \tau}$ for $i = 1, 2$. Clearly, $\int_0^\infty \gamma_i e^{-\gamma_i \tau} d\tau = 1$. Also, we consider the following new variables:

$$
\begin{cases}
Y(x,t) &= \int_0^\infty e^{-(\alpha_1 + \gamma_1)\tau} \left[ \dfrac{\beta_1 T(x, t - \tau) V(x, t - \tau)}{1 + \epsilon_1 V(x, t - \tau)} + \dfrac{\beta_2 T(x, t - \tau) I(x, t - \tau)}{1 + \epsilon_2 I(x, t - \tau)} \right] d\tau, \\
U(x,t) &= \int_0^\infty e^{-(\alpha_2 + \gamma_2)\tau} I(x, t - \tau) d\tau.
\end{cases}
$$

Then the variables $T, Y, I, U, V$ and $Z$ satisfy the following system:

$$
\begin{cases}
\dfrac{\partial T}{\partial t} &= d_T \triangle T + \lambda - dT(x, t) - \dfrac{\beta_1 T(x,t) V(x,t)}{1 + \epsilon_1 V(x,t)} - \dfrac{\beta_2 T(x,t) I(x,t)}{1 + \epsilon_2 I(x,t)}, \\
\dfrac{\partial Y}{\partial t} &= \dfrac{\beta_1 T(x,t) V(x,t)}{1 + \epsilon_1 V(x,t)} + \dfrac{\beta_2 T(x,t) I(x,t)}{1 + \epsilon_2 I(x,t)} - (\alpha_1 + \gamma_1) Y(x,t), \\
\dfrac{\partial I}{\partial t} &= d_I \triangle I + \gamma_1 Y(x,t) - aI(x,t) - pI(x,t)Z(x,t), \\
\dfrac{\partial U}{\partial t} &= I(x,t) - (\alpha_2 + \gamma_2) U(x,t), \\
\dfrac{\partial V}{\partial t} &= d_V \triangle V + k\gamma_2 U(x,t) - \mu V(x,t), \\
\dfrac{\partial Z}{\partial t} &= d_Z \triangle Z + cI(x,t)Z(x,t) - bZ(x,t).
\end{cases}
\tag{24}
$$

The threshold parameters $\mathcal{R}_0$ and $\mathcal{R}_1^Z$ for (24) are given by (11) and (15) with $\eta_1 = \dfrac{\gamma_1}{\alpha_1 + \gamma_1}$ and $\eta_2 = \dfrac{\gamma_2}{\alpha_2 + \gamma_2}$. For the simplicity of numerical illustrations, we consider one-dimensional bounded spatial domain $\Omega = [0, 10]$ with $d_T = d_I = d_Z = 0.01$ and $d_V = 0.02$. Also, we consider $\beta_2$ and $c$ as free parameters. All other parameter values are mentioned in Table 1.

**Table 1.** List of parameters and their values used in numerical simulations.

| Parameter | Value | Parameter | Value |
|:---:|:---:|:---:|:---:|
| $\lambda$ | 10 | $\alpha_2$ | 0.01 |
| $d$ | 0.0139 | $\gamma_2$ | 0.1 |
| $\beta_1$ | $2.4 \times 10^{-5}$ | $k$ | 50 |
| $a$ | 0.29 | $b$ | 0.1 |
| $\epsilon_1$ | 0.05 | $p$ | 0.01 |
| $\epsilon_2$ | 0.07 | $\mu$ | 3 |
| $\gamma_1$ | 0.1 | $\beta_2$ | Varied |
| $\alpha_1$ | 0.01 | $c$ | Varied |

When $\beta_2 = 1.5 \times 10^{-5}$ and $c = 0.02$, we have $\mathcal{R}_0 = 0.8539$. By the first result given in Corollary 1, the infection-free equilibrium $E_0(719.4245, 0, 0, 0)$ is globally asymptotically stable. This means that the virus is cleared, the infection dies out and the patient will be completely cured (see Figure 1).

When $\beta_2 = 5.6 \times 10^{-4}$ and $c = 0.02$, we obtained $\mathcal{R}_0 = 2.0830$ and $\mathcal{R}_1^Z = 0.8441$. From Corollary 1 2(i), we know that $E_1(620.0592, 4.2205, 65.6663, 0)$ is globally asymptotically stable (see Figure 2).

When $\beta_2 = 5.6 \times 10^{-4}$ and $c = 0.03$, we obtained $\mathcal{R}_0 = 2.0830$ and $\mathcal{R}_1^Z = 1.2662$. It follows from Corollary 1 2(ii) that $E_2(634.6036, 3.3333; 50.5051, 31.3883)$ is globally asymptotically stable (see Figure 3).

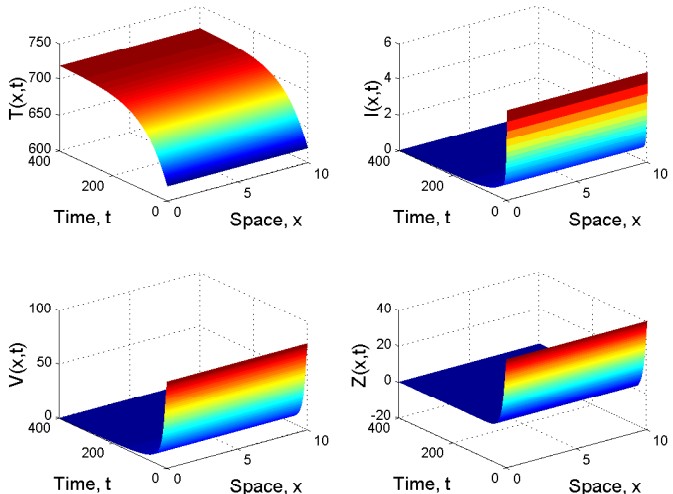

**Figure 1.** Spatiotemporal dynamics of the model (21) when $R_0 = 0.8539 \leq 1$.

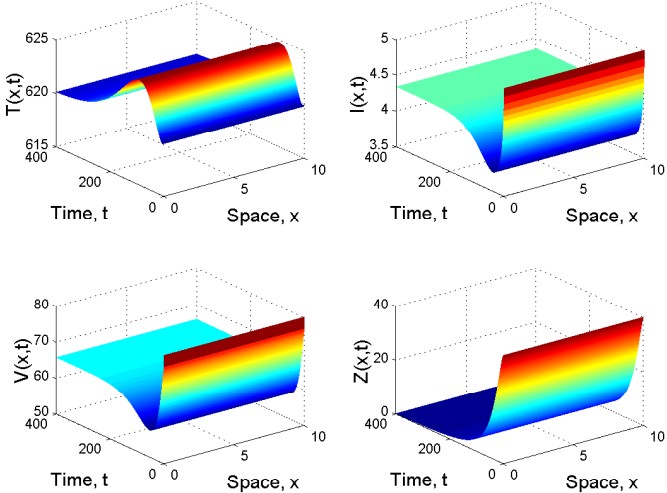

**Figure 2.** Spatiotemporal dynamics of the model (21) when $\mathcal{R}_0 = 2.0830 > 1$ and $\mathcal{R}_1^Z = 0.8441 \leq 1$.

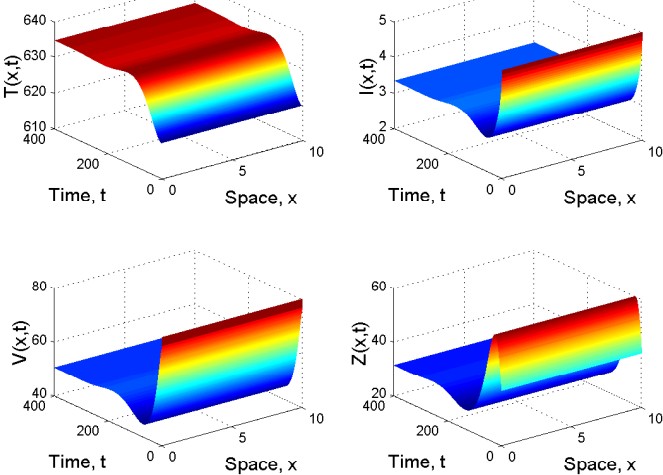

**Figure 3.** Spatiotemporal dynamics of the model (21) when $\mathcal{R}_0 = 2.0830 > 1$ and $\mathcal{R}_1^Z = 1.2662 > 1$.

## 5. Conclusions

In this article, we have proposed and investigated a generalized viral infection model with two infinite distributed delays, CTL immune response and spatial diffusion in both cells and virus. Also, the proposed model incorporated the classical virus-to-cell infection and the direct cell-to-cell transmission. Both routes of infection are modeled by two general incidence functions. Under some assumptions on these incidence functions, we have shown that the global dynamics of the model is completely determined by two threshold parameters that are the basic reproduction number $\mathcal{R}_0$ and the reproduction numbers for cellular immunity $\mathcal{R}_1^Z$. From the viewpoint of biology, we have proved that when $\mathcal{R}_0 \leq 1$ the infection-free equilibrium is globally asymptotically stable, which means that the virus is cleared and the infection dies out. Whereas, the virus persists in the host if $\mathcal{R}_0 > 1$ and two steady states appear, one without cellular immunity which is globally asymptotically stable if $\mathcal{R}_1^Z \leq 1$ and the other with cellular immunity which is globally asymptotically stable if $\mathcal{R}_1^Z > 1$. Hence, the activation of the CTL immune response is unable to eliminate the virus in vivo, but plays a fundamental role in the reduction of virus particles and infected cells. This last biological result can be easily deduced by comparing the components of virus particles and infected cells before and after the activation of cellular immunity. Since $\mathcal{R}_0$ and $\mathcal{R}_1^Z$ have no relation to the diffusion coefficients $d_T$, $d_I$, $d_V$ and $d_Z$, we conclude that the diffusion of cells and virus has no effect on the global stability of the three steady states of our PDE model with Neumann homogeneous boundary conditions. On the other hand, we have extended the models with ODEs [1,4,5], with DDEs [6–9] and with PDEs [17–19]. Moreover, the more recent works presented in [30,31] are improved and generalized.

**Funding:** This research received no external funding.

**Acknowledgments:** The author would like to express his sincere thanks to the editor and anonymous reviewers for their helpful comments and valuable suggestions which significantly improved the quality of this paper.

**Conflicts of Interest:** The author declares no conflict of interest.

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
