# Peer review of "Spatiotemporal Dynamics of a Generalized Viral Infection Model with Distributed Delays and CTL Immune Response"

_computation, doi:10.3390/computation7020021_

Round 1
Reviewer 1 Report
The function f depends on T,I,V, while g only on T, I. Comment on this please and give the biological or mathematicala motivation of this choice.
Complex formulas for dL1/dt, dL2/dt in the proof of Theorem 4 better to put in the appendix.
Author Response
Response to Reviewer 1
I thank the reviewer for the useful comments on my manuscript. A response
to each of the comments is given below.
1. The function f depends on T,I,V, while g only on T, I. Comment on this
please and give the biological or mathematical motivation of this choice.
Response: I fixed it by adding the biological meanings of hypotheses
on the incidence functions f(T, I, V ) and g(T, I). Also, I choose the
function f that depends on T,I,V, to include the standard incidence rate
of the form TV
T+I in order to investigate the dynamics of HBV infection.
2. Complex formulas for dL1/dt, dL2/dt in the proof of Theorem 4 better
to put in the appendix.
Response: Thank you for this remark, but the sentences in the proof
are interconnected.
Reviewer 2 Report
The manuscript describes a generalization of previous viral dynamics models that include a CTL response. While the model is formulated as PDEs that include diffusion of both cells and virus, the analysis does not include spatial dynamics, so some of the claims of the manuscript are a bit of an over-reach. However, the analysis of the temporal dynamics is sound, so the paper can be accepted after some major revisions.
Specific comments:
The title of the manuscript is misleading. As mentioned above, the model is formulated as PDEs, but the analysis only examines the spatially uniform steady states (and the author does not even mention that these are spatially uniform states). Thus, the manuscript title should not start with "Spatiotemporal dynamics" since spatiotemporal dynamics are not examined at all. The author can either remove the PDE model and simply keep the ODE model that is actually analyzed, or actually perform an analysis of the spatiotemporal dynamics of the system by either doing simulations or examining traveling wave solutions to the PDE model.
It's not clear why the proliferation of CTLs depends on both the number of CTLs and the number of infectious cells. Proliferation of cells usually depends only on the number of cells of the same type.
On page 2, the author states "From biological viewpoint, the above hypotheses are reasonable and consistent with the reality." This statement needs explanation. It's not immediately clear why the hypotheses are biologically required.
It is not clear where hypothesis 4 comes from or why it should be true.
There are a number of places where there are some English grammar problems (see attached file).

Author Response
Response to Reviewer 2
I thank the reviewer for the useful comments on my manuscript. A response to each of the comments is given below.
The manuscript describes a generalization of previous viral dynamics models that include a CTL response. While the model is formulated as PDEs that include diffusion of both cells and virus, the analysis does not include spatial dynamics, so some of the claims of the manuscript are a bit of an over-reach. However, the analysis of the temporal dynamics is sound, so the paper can be accepted after some major revisions.
—Reply: Thank you for your positive comments on the manuscript and encouraging words on the quality of the paper.
1. The title of the manuscript is misleading. As mentioned above, the model is formulated as PDEs, but the analysis only examines the spatially uniform steady states (and the author does not even mention that these are spatially uniform states). Thus, the manuscript title should not start with ”Spatiotemporal dynamics” since spatiotemporal dynamics are not examined at all. The author can either remove the PDE model and simply keep the ODE model that is actually analyzed, or actually perform an analysis of the spatiotemporal dynamics of the system by either doing simulations or examining traveling wave solutions to the PDE model.
Response: I added the numerical simulations of the PDE model (see the revised version). For the traveling wave solutions, I will investigate it in my future work.
2. It’s not clear why the proliferation of CTLs depends on both the number of CTLs and the number of infectious cells. Proliferation of cells usually depends only on the number of cells of the same type.
Response: I fixed it by adding ”CTL cells expand in response to viral antigens derived from infected cells at rate cIZ”.
3. On page 2, the author states ”From biological viewpoint, the above hypotheses are reasonable and consistent with the reality.” This statement needs explanation. It’s not immediately clear why the hypotheses are biologically required.
Response: I explained it (see the revised version).
4. It is not clear where hypothesis 4 comes from or why it should be true.
Response: I fixed it (see Remark in the revised version).
5. There are a number of places where there are some English grammar problems (see attached file).
Response: I fixed it. Thank you for your attentive reading.
Round 2
Reviewer 2 Report
The authors have addressed my comments. I now recommend the manuscript be published.